# The Effect of Intracellular Calcium Buffer Bapta on Epileptiform Activity of Hippocampal Neurons

**DOI:** 10.3390/ijms26157596

**Published:** 2025-08-06

**Authors:** V. P. Zinchenko, I. Yu. Teplov, F. V. Tyurin, A. E. Malibayeva, B. K. Kairat, S. T. Tuleukhanov

**Affiliations:** 1Federal Research Center “Pushchino Scientific Center for Biological Research of the Russian Academy of Sciences”, Institute of Cell Biophysics of the Russian Academy of Sciences, Pushchino 142290, Russia; teplov_iy@pbsras.rus (I.Y.T.); tiurin_teodor@pbsras.ru (F.V.T.); 2Faculty of Biology and Biotechnology, Al-Farabi Kazakh National University, Almaty 050040, Kazakhstan; arailym.malibayeva@kaznu.edu.kz (A.E.M.); bakytzhan.kairat@kaznu.edu.kz (B.K.K.); sultan.tuleuhanov@kaznu.edu.kz (S.T.T.)

**Keywords:** intracellular calcium buffer, BAPTA, paroxysmal depolarizing shift (PDS), epileptiform neuronal activity

## Abstract

The rhythm of epileptiform activity occurs in various brain injuries (ischemia, stroke, concussion, mechanical damage, AD, PD). The epileptiform rhythm is accompanied by periodic Ca^2+^ pulses, which are necessary for the neurotransmitter release, the repair of damaged connections between neurons, and the growth of new projections. The duration and amplitude of these pulses depend on intracellular calcium-binding proteins. The effect of the synthetic fast calcium buffer BAPTA on the epileptiform activity of neurons induced by the GABA(A)-receptor inhibitor, bicuculline, was investigated in a 14-DIV rat hippocampal culture. In the epileptiform activity mode, neurons periodically synchronously generate action potential (AP) bursts in the form of paroxysmal depolarization shift (PDS) clusters and their corresponding high-amplitude Ca^2+^ pulses. Changes in the paroxysmal activity and Ca^2+^ pulses were recorded continuously for 10–11 min as BAPTA accumulated. It was shown that during BAPTA accumulation, transformation of neuronal patch activity occurs. Moreover, GABAergic and glutamatergic neurons respond differently to the presence of calcium buffer. Experiments were performed on two populations of neurons: a population of GABAergic neurons that responded selectively to ATPA, a calcium-permeable GluK1 kainate receptor agonist, and a population of glutamatergic neurons with a large amplitude of cluster depolarization (greater than −20 mV). These neurons made up the majority of neurons. In the population of GABAergic neurons, during BAPTA accumulation, the amplitude of PDS clusters decreases, which leads to a switch from the PDS mode to the classical burst mode with an increase in the electrical activity of the neuron. In glutamatergic neurons, the duration of PDS clusters decreased during BAPTA accumulation. However, the amplitude changed little. The data obtained showed that endogenous calcium-binding proteins play a significant role in switching the epileptiform rhythm to the recovery rhythm and perform a neuroprotective function by reducing the duration of impulses in excitatory neurons and the amplitude of impulses in inhibitory neurons.

## 1. Introduction

Generally, the epileptiform activity of brain neurons refers to the rhythm of interictal brain activity that is observed between epileptic seizures. In epileptiform neuronal activity, the normal bursting electrical activity is transformed into a paroxysmal depolarizing shift (PDS) cluster activity [1]. PDS clusters have a complex structure that contains in addition to two oscillation systems (fast oscillations of AP and slow oscillations of AP bursts), a third, little studied oscillation system, PDS. All three oscillatory systems include a set of potential- and ligand-dependent ion channels, the sequential opening and closing of which forms such a complex electrical signal—the PDS cluster. Importantly, this type of brain neuron activity has been observed not only in epilepsy but also in stroke, Alzheimer’s disease (AD), Parkinson’s disease (PD) [2,3], mechanical damage, concussion, and early periods of brain development [4,5].

According to the World Health Organization (WHO), tens of millions of people suffer from various neurological disorders worldwide. Approximately 50–70 million people have epilepsy [6]. Parkinson’s disease affects approximately 1% of the population over the age of 60 years, and grows to 4% at the age of 80 years [7,8]. Sixty-nine million individuals worldwide are estimated to sustain a traumatic brain injury each year [9]. Stroke is the third highest cause of morbidity and mortality in many countries of the world, following ischemic heart disease and malignant diseases [10]. Approximately 1% of the global population (50–70 million people) have epilepsy [11]. It has been suggested that rhythm is necessary to repair damage and establish new connections between neurons [12]. Understanding the mechanisms underlying changes in the amplitude and frequency of the rhythm can help us control the aforementioned processes. Therefore, the current task is not only to suppress hyperactivity in pathologies, but to switch the rhythm of epileptiform activity into the rhythm of the development/restoration of damaged contacts.

Ca^2+^ ions regulate many processes in the cell and are one of the main regulators of neuronal bursting activity. They do this both directly by binding to targets and indirectly by binding to Ca^2+^-binding proteins: parvalbumin, calmodulin, S100 proteins and calcineurin, calreticulin, and calsequestrin [13,14]. It has been shown that the generation of AP bursts during epileptiform activity is accompanied by Ca^2+^ impulses [15]. It is known that the amplitude and shape of calcium impulses depend on the concentration of intracellular calcium-binding proteins (CBPs), which can delay or eliminate the effect of Ca^2+^ on transmitter release or on excitation [16].

Calretinin and calbindin belong to fast Ca^2+^-binding proteins [17], and are found in some subtypes of neurons in buffer concentrations [18]. The action of such CBPs can be mimicked by using a synthetic analog, BAPTA [19]. The esterified form of BAPTA-AM added to the cell incubation medium slowly (within 10 min) accumulates in cells, where the ester bonds are hydrolyzed by intracellular esterases, gradually increasing the Ca^2+^ buffer concentration. Since K_d_ for Ca^2+^ = 0.2 μM, the presence of BAPTA does not change the steady-state values of cytosolic Ca^2+^ concentration ([Ca^2+^]_i_), but may reduce the amplitude and smooth out Ca^2+^ pulses during epileptiform activity of the neuron. The concentration of BAPTA-AM was selected experimentally. In other experiments, BAPTA was used at concentrations of 10, 25, and 50 μM. Lower concentrations did not affect the pulse parameters, while higher concentrations caused rapid inhibition of pulsations. The relationship between the dose and the effect of intracellular BAPTA is evident when BAPTA is accumulated during the experiment.

Calcium currents, unlike sodium and potassium currents, significantly change the [Ca^2+^]_i_ during the PDS clusters generation, which is necessary for the neurotransmitters release. However, the role of Ca^2+^ concentration and CBP in the regulation of PDS clusters and in rhythmogenesis in different types of neurons during epileptiform activity is poorly understood.

## 2. Results

### 2.1. Selection of Neurons. ATPA-Sensitive GABAergic Neurons

It was previously shown that in the mode of epileptiform activity induced by the GABA(A)-receptor inhibitor bicuculline in hippocampal cell culture, neurons generate bursts of AP in the form of PDS clusters, which differ in amplitude and frequency [20].

We investigated the effect of BAPTA on two groups of neurons. We chose glutamatergic neurons with a large cluster depolarization amplitude (greater than −20 mV), in which all Na^+^-channels were inactivated after the first AP. Such neurons constituted the majority in the culture (Figure 1A).

Among the inhibitory neurons, we selected neurons that responded specifically to the (RS)-2-amino-3-(3-hydroxy-5-tert-butylisoxazol-4-yl) propanoic acid (ATPA), a selective agonist of the GLU(K5) receptor subtype (Figure 1B). Previous studies have shown that these neurons are GABAergic and express calcium-permeable kainate receptors (CP-KARs) [21,22,23].

A standard experiment to identify ATPA-sensitive neurons is shown in Figure 2A. In an initially spontaneously pulsed hippocampal neuronal culture, a near-maximal dose of ATPA caused a [Ca^2+^]_i_ increase in GABAergic neurons (highlighted in red). Glutamatergic neurons are indicated by other blue-green-gray colors. ATPA-induced elevation of [Ca^2+^]_i_ causes GABA release [24] and inhibits [Ca^2+^]_i_ pulses in innervated glutamate neurons (Figure 2B). The subsequent addition of bicuculline largely reverses the effects of GABA, inducing epileptiform activity in the form of synchronized PDS cluster oscillations in all neurons. The amplitude of impulses in most neurons increases compared to the control before supplementation. BAPTA increases the amplitude of [Ca^2+^]i oscillations in glutamatergic neurons (gray curves) and decreases them in GABAergic neurons (Figure 2A).

### 2.2. The Effect of BAPTA on PDS Clusters During Epileptiform Neuronal Activity

The effect of BAPTA was studied in glutamate and GABAergic neurons, which were selected based on the above-mentioned characteristics. The resting membrane potential of GABAergic and glutamatergic neurons did not differ significantly and averaged −65 mV. The membrane potential at which the Na^+^ current was generated in GABAergic neurons was −55 mV on average and required a small depolarization to activate it. In contrast, glutamatergic neurons were activated at an average of −42 mV and required a strong depolarization. Figure 3 and Figure 4 show changes in the membrane potentials of two neurons during bicuculline-induced epileptiform activity, both in the control and after the addition of 30 µM BAPTA-AM. BAPTA-AM was added at 270 s. In both cases, BAPTA-AM caused an increase in the frequency of AP bursts, from 0.08–0.1 Hz to 0.2–0.3 Hz on average. In both types of neurons, a small depolarization (10–12 mV) is observed during BAPTA accumulation (Figure 3A and Figure 4A). The amplitude of the first AP in each cluster does not change, indicating that the cell membrane is intact and Na^+^ gradients are maintained. Since the depolarization effect occurs as BAPTA accumulates rather than immediately, inhibition occurs on the inner side of the membrane. The figures also show a gradual increase in the frequency of AP bursts. If slow depolarization results from K^+^ channel inhibition by BAPTA-AM, then its effect should be washed off, as illustrated in Figure 4A. The effect of the increased frequency of clusters is also washed away. The slow depolarization that grows as BAPTA accumulates appears to be the result of BAPTA-AM inhibiting the K^+^ channel [25,26], since the effect is washed out in both potential and frequency.

Figure 3B–F and Figure 4B–F illustrate the changes in PDS cluster structure in glutamatergic and GABAergic neurons, respectively, as BAPTA accumulates. In the glutamate neuron, the cluster amplitude exceeds −20 mV (see Figure 1A). A single action potential (AP) is generated at the leading edge of the PDS. The PDS cluster duration decreases as BAPTA accumulates, from 9 PDS to 1 PDS (from 1.43 s to 0.15 s), but the amplitude remains unchanged. After that, the pulsations will stop.

Figure 4B–E show changes in PDS clusters during BAPTA accumulation in a selected GABAergic neuron. Figure 4B shows the control PDS cluster taken from Figure 4A, which was recorded before BAPTA addition. After incubation with BAPTA-AM only 2–3 min (Figure 4C), there is a sharp decrease in the PDS cluster amplitude and a slowing of the leading edge of the burst. This results in a switch in PDS cluster mode to burst mode, which exhibits the classic inverse relationship between AP amplitude and slow depolarization amplitude (Figure 4D). After washing, the effect of BAPTA-AM was maintained (Figure 4E,F).

A comparison of PDS clusters in the presence and absence of BAPTA in a glutamatergic neuron (Figure 5) showed that cluster durations and their associated APs were maximally suppressed (red curve). Previous studies have shown that Kv7-type potassium channels are involved in regulating cluster duration [27]. Figure 6 shows that the Kv7 activator retigabine decreases cluster duration without significantly affecting amplitude. It is likely that BAPTA’s effect is also due to Kv7 activation. Kv7 activity is known to be inhibited by CaCaM [28,29], which appears to be impaired in the presence of BAPTA.

Thus, the main effect of the intracellular fast calcium buffer in glutamate neurons is to reduce the duration of the PDS cluster. In contrast, in GABAergic neurons, the calcium buffer promotes a shift in cluster activity to classical AP bursts by reducing cluster amplitude. These mechanisms both explain the neuroprotective effect of intracellular Ca^2+^ buffers and may contribute to the transition from epileptiform to developmental rhythms.

## 3. Discussion

This study demonstrates that the fast-binding calcium buffer plays an active role in forming AP burst patterns and regulates the rhythm of neuronal burst activity. It does so by regulating the amplitude and duration of PDS clusters and calcium pulses. The clusters’ frequency and duration are not constant in the control group and fluctuate chaotically around a certain value. This behavior suggests the involvement of a stochastic process in both impulse generation and cluster duration regulation. This process involves the generation and addition of hundreds of elementary postsynaptic impulses. Typically, neighboring clusters differ in duration by 1 PDS.

Figure 3A and Figure 4A demonstrate that BAPTA-AM increases the oscillation frequency, corresponding to depolarization-dependent neuronal excitation. As previously demonstrated, this effect may be related to BAPTA-AM’s inhibition of the potassium channel [30]. Since this effect develops over time, the inhibition appears to occur on the inner side of the membrane.

In glutamatergic neurons, BAPTA rapidly decreased the duration of the PDS cluster without significantly affecting amplitude. This helps maintain the rhythm within a safe range of other parameters, such as frequencies. Reducing cluster duration is critical because it determines the duration of the Ca^2+^ pulse. This reduction is observed in the action of many neuroprotective compounds [22,31].

The pulse duration is determined by the lifetime of the control potassium and calcium channels. However, Ca^2+^ channels also alter the amplitude of the signal. The mechanism by which signal duration decreases without amplitude suppression may be related to the activation of low-threshold Kv7-type potassium channels [32], whose activity is controlled by PIP2 [33,34], the βγ subunit, CaCaM, and phosphorylation [35]. Since an increase in either of the first two is unlikely in the presence of BAPTA, activation by decreasing CaCaM seems to occur in this case.

Unlike glutamatergic neurons, the signal duration of GABAergic neurons decreased insignificantly during BAPTA accumulation. However, the amplitude of PDS clusters decreased significantly (Figure 4B–F). This resulted in a switch to the classical burst mode, which increased the electrical activity of the neuron (i.e., increased AP amplitude). This latter effect appears to be close to the physiological effect, as the presence of fast CBPs in GABAergic neurons has been demonstrated and is even a marker for them [36]. The effect is similar to that of AP-5, an NMDAR inhibitor that reduces the depolarizing current across the postsynaptic membrane [1]. It is also shown there that a decrease in the amplitude of PDS clusters is accompanied by a switch to classical burst activity and an increase in AP amplitude. In our case, Ca^2+^ currents across the membrane can only decrease through attenuation of the calcium-dependent Ca^2+^ release from the ER, which triggers the depolarizing current. The steepness of the leading edge of the clusters is proportional to the Ca^2+^ current and decreases in the presence of BAPTA in both types of neurons.

Most of the experimental data indicates that calcium channels and calcium ions are among the main regulators of PDS clusters [15,22,37,38]. These channels participate in generating both low-amplitude, slow depolarizing pulses and high-amplitude depolarizing pulses that induce neurotransmitter release. Furthermore, calcium currents and [Ca^2+^]_i_ control the opposing processes of depolarization and hyperpolarization, respectively. The regulation of bursting activity (including PDS clusters) by Ca^2+^ ions is assumed to target both excitation and inhibition. Accordingly, it consists of two processes: the depolarizing (excitatory) effect of Ca^2+^ currents occurs through the opening of various Ca^2+^ channels in the plasma membrane and synapses; the hyperpolarizing (inhibitory) effect occurs through the opening of calcium-dependent K^+^ and Cl^−^ channels and the activation of GABA release. Clearly, the inhibitory effect occurs with a delay relative to the excitatory, as it takes time to change [Ca^2+^]_i_. Thus, for the first time, it has been shown that an intracellular fast Ca^2+^ buffer suppresses the epileptiform activity of neurons. Intracellular fast Ca^2+^ buffer decreases the duration of AP bursts (PDS clusters) in glutamate neurons and reduces the amplitude of PDS clusters in GABAergic neurons. The decrease in AP burst duration (PDS cluster duration) in glutamate neurons is assumed to result from the activation of low-threshold, slow-inactivating, voltage-dependent Kv7 channels [32]. Decreasing the duration of PDS clusters in glutamate neurons and decreasing the amplitude of PDS clusters in GABAergic neurons may contribute to generating a recovery rhythm.

In the future, we aim to control the rhythm of neural network development and recovery after damage. This work shows that intracellular fast Ca^2+^ binding proteins can be involved in this process. We will use the filling of Ca^2+^ buffers in inhibitory neurons to switch the epileptic rhythm into the rhythm of neural network recovery after brain injury.

## 4. Materials and Methods

### 4.1. Cell Culture 

In the present experiments, epileptiform activity was induced in hippocampal cell culture by the GABA(A) receptor inhibitor bicuculline. Neuron-glial cultures were derived from the hippocampi of newborn Wistar rats (P0–2), as described earlier [21,24,39]. Wistar pups (P0–2) of both sexes were euthanized with deep-inhaled anesthesia and decapitated. (Original source—animal facility of The Branch of the IBCh RAS in Pushchino; RRID: Not registered). Cultures were grown in a CO_2_ incubator at 37 °C and 95% humidity for two weeks and then were used in experiments. We used 12–14 DIV cultures in all experiments. The concentration of BAPTA AM was selected experimentally. In other experiments, BAPTA was used at concentrations of 10, 25, and 50 μM. Lower concentrations did not affect the pulse parameters, while higher concentrations caused rapid inhibition of pulsations. The relationship between the dose and the effect of intracellular BAPTA is evident when BAPTA is accumulated during the experiment.

### 4.2. Fluorescent [Ca^2+^]_i_ Measurements 

Hippocampal cell cultures were stained with Fluo-8/AM (AAT Bioquest, Inc. (Pleasanton, CA, USA)/FluoriCa-8 AM (Lumiprobe RUS, Moscow, Russia), dissolved in Hank’s Balanced Salt Solution (HBSS), consisting of (mM): 136 NaCl, 3 KCl, 0.8 MgSO_4_, 1.25 KH_2_PO_4_, 0.35 Na_2_HPO_4_, 1.4 CaCl_2_, 10 glucose, and 10 HEPES; pH 7.35. The cells were incubated with the probe at 28–30 °C for 40 min (working concentration 3 µM). The neurons were subsequently washed with HBSS and incubated for a minimum of 10 min in fresh culture medium. The excitation wavelength for Fluo-8/AM was 490 nm, and the emission wavelength was 525 nm. All imaging experiments were performed at 28–30 °C. The fluctuations in [Ca^2+^]_i_ were recorded using an inverted motorized microscope Leica DMI6000B (Leica Microsystems, Wetzlar, Germany), equipped with a high-speed monochrome CCD camera HAMAMATSU C9100 (Hamamatsu Photonics K.K., Hamamatsu City, Japan) and a high-speed excitation filter wheel system from Leica’s Ultra-Fast Filter Wheels. A Leica HC PL APO 20×/0.7 IMM objective (Leica Microsystems, Wetzlar, Germany) was used. The excitation source for fluorescence was a Leica EL6000 illuminator with a high-pressure mercury lamp HBO 103 W/2 (Leica Microsystems, Wetzlar, Germany). For fluorescence measurement, a round coverslip with the cell culture stained with Fluo-8 was mounted in a special measurement chamber. The volume of HBSS in the chamber was 1 mL. Reagent additions and washouts were performed by changing the medium to a five-fold volume using a system that provided perfusion at a rate of 15 mL/min.

The fluorescence intensity was collected from the soma during the analysis of the time-lapse series. The mean background fluorescence for each series of images was obtained by averaging the signals collected from 10 ROIs set on the areas of the culture without soma and processes of the cells, and the obtained mean background fluorescence kinetics (baselines) were subtracted from the corresponding signal of each analyzed cell. Changes in [Ca^2+^]_i_ are presented as (ΔF/F) and were calculated in ImageJ (NIH, Bethesda, MD, USA, https://imagej.net/ij/) programming environment to generate single-neuron activity traces. The results are presented as representative recordings of calcium signals in individual cells.

### 4.3. Electrophysiological Measurements 

Experiments using the patch-clamp technique in whole-cell configuration were carried out on fluorescent station with built-in microincubator and electrophysiological patch-clamp setup equipped with a Hamamatsu ORCA-Flash camera. Electrophysiological characteristics of neurons were recorded at 28 °C using an Axopatch 200B amplifier (Axon Instruments, Union City, CA, USA). Data were digitized using a low-noise data acquisition system (Axon DigiData 1440A digital converter) (Molecular Devices, San Jose, CA, USA) with pCLAMP 10 software from Axon Instruments (USA). Glass capillaries made from borosilicate glass (outer diameter 1.5 mm, inner diameter 0.06 mm, length 10 cm, manufactured by Sutter Instrument, Novato, CA, USA) were used to make micropipettes. Pipettes were pulled using a Narishige PC-100 puller (Tokyo, Japan). The resistance of the pipette tip was 4–5 MΩ. The internal pipette solution had the following composition (mM): 5 KCl, 130 potassium gluconate, 1 MgCl_2_ × 6H_2_O, 0.25 EGTA, 4 HEPES, 2 Na2-ATP, 0.3 Mg-ATP, 0.3 Na-GTP, and 10 Na2-phosphocreatine (305–310 mOsm, pH 7.2). The coverslip with the culture was placed in an experimental chamber containing Hank’s solution (volume of 1 mL). Reagent additions and washouts were performed using a perfusion system similar to the method described above.

### 4.4. Reagents 

The following reagents were used in the experiments: (−)-bicuculline methochloride, ATPA, retigabine (Tocris Bioscience, Bristol, UK), Neurobasal-A medium and B-27 supplement (Thermofisher, Waltham, MA, USA), trypsin (1%) (Life Technologies, Grand Island, NY, USA); and BAPTA-AM (Sigma-Aldrich, Saint Louis, MO, USA), Neurobasal medium (H333), and penicilin-streptomycin (A063) Paneco, Moscow, Russian Federation.

Statistical and data analysis ImageJ software (National Institutes of Health, Bethesda, MD, USA) was used for image analysis. Origin Pro 2021 version 9.8.0.200 was used for graph creation and analysis (OriginLab, Northampton, MA, USA). Electrophysiological data were analyzed using ClampFit 10 software (Molecular Devices, San Jose, CA, USA). All experiments were performed using the cultures from at least 2 to 3 different animals.

## Figures and Tables

**Figure 1 ijms-26-07596-f001:**
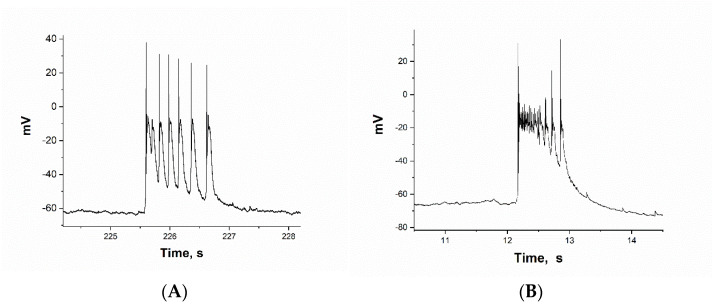
(**A**) PDS cluster of glutamatergic neurons. (**B**) PDS cluster of GABAergic neurons.

**Figure 2 ijms-26-07596-f002:**
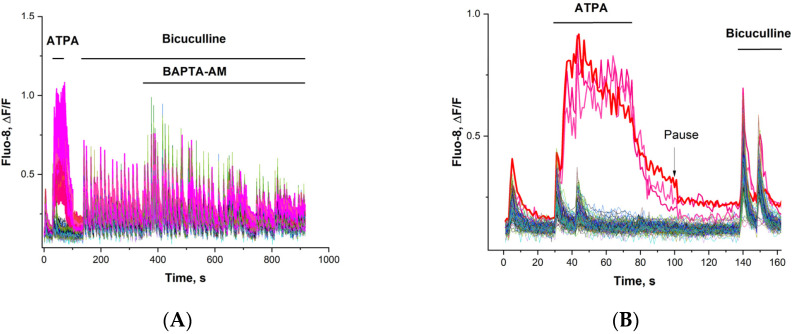
(**A**) Spontaneously pulsing neurons. [Ca^2+^]_i_ changes in GABAergic neurons (highlighted in red) and in glutamatergic neurons (highlighted in blue-green-gray) upon successive addition of ATPA (500 nM), bicuculline (20 µM), and BAPTA-AM (30 µM). In this experiment, 60 out of 210 neurons in the field of view responded to ATPA. (**B**) Initial part of the subfigure (**A**). ATPA causes a [Ca^2+^]_i_ increase in GABAergic neurons (3 of which are highlighted in red). ATPA inhibits [Ca^2+^]_i_ pulses in glutamate neurons. Pause 10 min.

**Figure 3 ijms-26-07596-f003:**
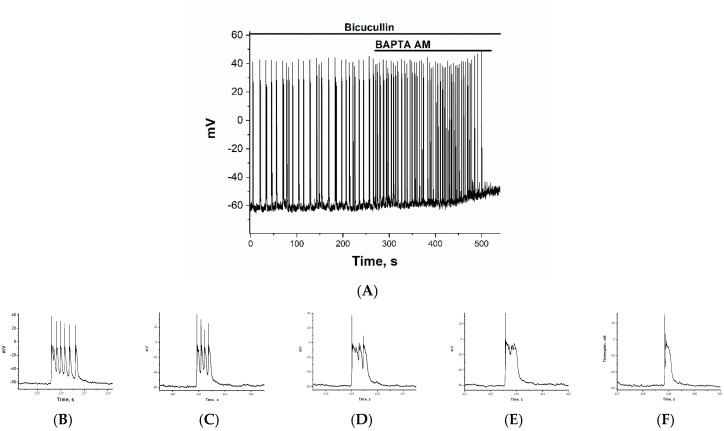
(**A**) Changes in the membrane potential of glutamatergic neurons in epileptiform activity mode in response to 30 μM BAPTA-AM added after 270 s. PDS cluster: (**B**) in control; (**C**) 120 s incubation with BAPTA; (**D**) 149 s incubation with BAPTA; (**E**) 152 s incubation with BAPTA; and (**F**) 178 s incubation with BAPTA.

**Figure 4 ijms-26-07596-f004:**
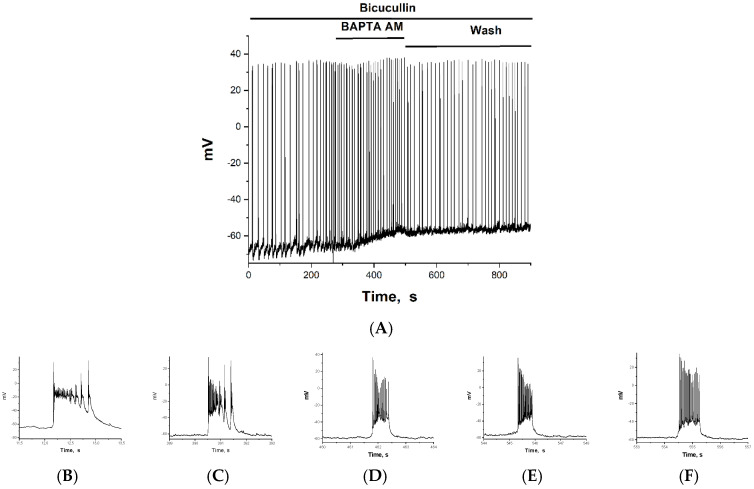
(**A**) Changes in the membrane potential of GABAergic neurons in epileptiform activity mode in response to 30 μM BAPTA-AM added after 270 s. PDS cluster: (**B**) in control; (**C**) 120 s incubation with BAPTA; (**D**) 211 s incubation with BAPTA; (**E**) 45 s after washing from BAPTA; and (**F**) 54 s after washing from BAPTA.

**Figure 5 ijms-26-07596-f005:**
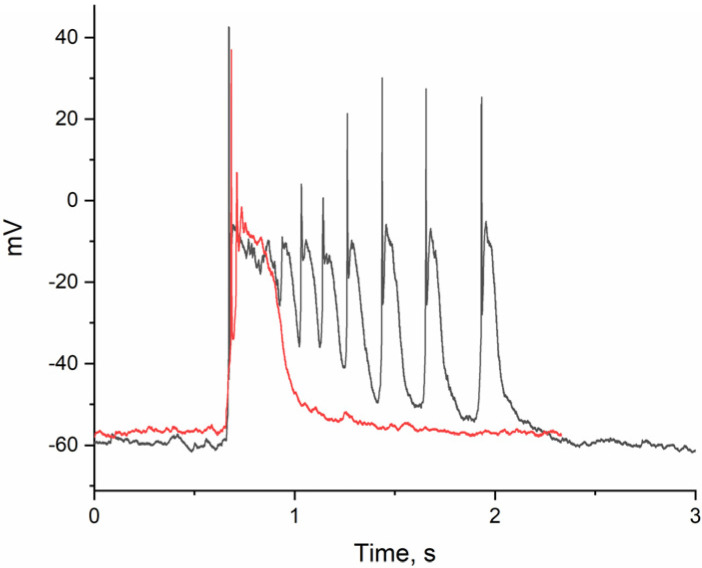
Comparison of PDS clusters in glutamatergic neuron in control (black curve) and after 3 min incubation with BAPTA-AM (red curve).

**Figure 6 ijms-26-07596-f006:**
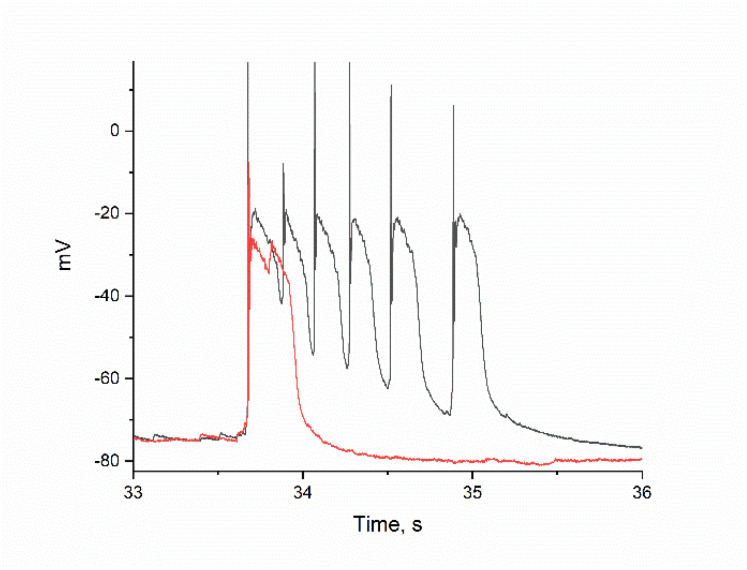
PDS cluster in glutamatergic neuron in control (black curve) and in the presence of Kv7 channel activator retigabine (2.5 μM) (red curve).

## Data Availability

The original contributions presented in this study are included in the article. Further inquiries can be directed to the corresponding author(s).

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
