# Peer review of "The Effect of Intracellular Calcium Buffer Bapta on Epileptiform Activity of Hippocampal Neurons"

_ijms, 2025, doi:10.3390/ijms26157596_

Round 1
Reviewer 1 Report
Comments and Suggestions for Authors
The manuscript titled “The Effect of Intracelluar Calcium Buffer Bapta on Epileptiform Activity of Hippocampal neurons” by Zinchenko, V.P.; et al. is a scientific work where the authors addressed the interplay of intracellular buffer concentrations on neurons of different nature (glutamatergic and GABAergic neurons). The most relevant outcomes found in this research could serve to have a more complete outlook and better understand the underlying mechanisms of the role of calcium in the regulation of neuronal activity with impact in many human brain malignancies. This is a topic of growing interest and furthermore, the manuscript is generally well-written.
However, it exists some points that need to be addressed (please, see them below detailed point-by-point) to improve the scientific quality of the submitted manuscript paper before this article will be consider for its publication in the International Journal of Molecular Sciences.
1) Introduction. “Generally, the term epileptiform activity of brain neurons (…) electrical activity is transformed into paraxysmal depolarizing shift (PDS) cluster (…) the brain neuron activity ahs been observed not only in epilepsy but also in stroke, Alzheimer’s disease (AD), Parkinson’s disease (PD) (…) early period of brain development” (lines 40-50). Could the authors provide quantitative data insights according to the worldwide global burdens of neuronological disorders as the above described? This will significantly aid the potential readers to better understand the significance of this devoted research.
2) “Ca2+ ions are one of the main regulators of neuronal bursting activity (…) generation of AP burst during epileptiform activity is accompanied by Ca2+ impulses. (…) Ca2+ on transmitter release or on excitation” (lines 56-60). Here, even if I agree with these statements provided by the authors, it should be also remarkable to highlight the importance of intracellular calcium concentration not only in the onset and development of neurological disorders triggering the fibril formation [1], but also the impact of environmental conditions inner the brain [2]. This will strengthen the gathered findings in this research.
[1] https://doi.org/10.3390/biom14091091
[2] https://doi.org/10.3390/ijms222212382
3) “It is known that the amplitude and shape of calcium impulsed depend on the concentration of intracellular calcium-binding proteins (CBPs), which can delay or eliminate the effect of Ca2+ (…) Calretinin and calbinding belong to fast Ca2+-binding proteins (…) The action of such CBPs can be mimicked by using a synthetic analog, BAPTA” (lines 58-63). Did the authors also test the effect of calmoduling on the calcium intracellular signaling processes? This information should be placed in the respective Results sections.
4) Results. Did the authors carry out electrophysiology experiments in absence of calcium as negative control (e.g. with other positive divalent cations as magnesium or in presence of quelators)? Some further insights need to be furnished in this regard.
5) “The mechanism (…) [Error! Reference source not found.]” (lines 168-172). It was an issue related to the bibliography citation manager software tool. The authors should fix it. This comment needs to be taken into account for the rest of the main mansucript body text.
6) Finally, did the authors monitor the resting membrane potential and imput neuronal resistance during the Patch Clamp experiments to ensure that the cells remain healthy and viable after the calcium-activated currents?
7) “Discussion” (lines 148-207). This section perfectly remarks the most relevant outcomes found by the authors in this work and also the promising future perspectives. It may be desirable to add a brief statement to discuss about the potential future action lines to pursue the topic covered in this research.
Author Response
1) Introduction. “Generally, the term epileptiform activity of brain neurons (…) electrical activity is transformed into paraxysmal depolarizing shift (PDS) cluster (…) the brain neuron activity has been observed not only in epilepsy but also in stroke, Alzheimer’s disease (AD), Parkinson’s disease (PD) (…) early period of brain development” (lines 40-50). Could the authors provide quantitative data insights according to the worldwide global burdens of neuronological disorders as the above described? This will significantly aid the potential readers to better understand the significance of this devoted research.
Response:
According to the World Health Organization (WHO) tens of millions of people suffer from various neurological disorders worldwide. Approximately 50–70 million people have epilepsy [Liu J, Zhang P, Zou Q, Liang J, Chen Y, Cai Y, Li S, Li J, Su J, Li Q. Status of epilepsy in the tropics: An overlooked perspective. Epilepsia Open. 2023 Mar;8(1):32-45]. Parkinson’s disease affecting approximately 1% of the population over the age of 60 years, growing to 4% at the age of 80 years. [Polymeropoulos MH, Higgins JJ, Golbe LI, Johnson WG, Ide SE, Di Iorio G, Sanges G, Stenroos ES, Pho LT, Schaffer AA, Lazzarini AM, Nussbaum RL, Duvoisin RC. Mapping of a gene for Parkinson's disease to chromosome 4q21-q23. Science. 1996 Nov 15;274(5290):1197-9.] [Zaltieri M, Longhena F, Pizzi M, Missale C, Spano P, Bellucci A. Mitochondrial Dysfunction and α-Synuclein Synaptic Pathology in Parkinson's Disease: Who's on First? Parkinsons Dis. 2015;2015:108029. doi: 10.1155/2015/108029. Epub 2015 Mar 31. ]. Sixty-nine million individuals worldwide are estimated to sustain a Traumatic brain injury each year. [Dewan MC, Rattani A, Gupta S, Baticulon RE, Hung YC, Punchak M, Agrawal A, Adeleye AO, Shrime MG, Rubiano AM, Rosenfeld JV, Park KB. Estimating the global incidence of traumatic brain injury. J Neurosurg. 2018 Apr 27;130(4):1080-1097]. Stroke is the third highest cause of morbidity and mortality in many countries of the world, following ischemic heart disease and malignant diseases. [Petrović G. Faktori rizika u pojavi cerebrovaskularnog insulta [Risk factors for development of cerebrovascular stroke]. Med Pregl. 2000 Mar-Apr;53(3-4):207-14. Croatian. PMID: 10965691.] Approximately 1% of the global population (50–70 million people) have epilepsy [Ngugi, A.K.; Bottomley, C.; Kleinschmidt, I.; Sander, J.W.; Newton, C.R. Estimation of the burden of active and life-time epilepsy: A meta-analytic approach. Epilepsia 2010, 51, 883–890].
Response: Added
2) “Ca2+ ions are one of the main regulators of neuronal bursting activity (…) generation of AP burst during epileptiform activity is accompanied by Ca2+ impulses. (…) Ca2+ on transmitter release or on excitation” (lines 56-60). Here, even if I agree with these statements provided by the authors, it should be also remarkable to highlight the importance of intracellular calcium concentration not only in the onset and development of neurological disorders triggering the fibril formation [1], but also the impact of environmental conditions inner the brain [2]. This will strengthen the gathered findings in this research.
Response: Ca2+ ions regulate many processes in the cell. They do this both directly by binding to targets and indirectly by binding to Ca2+-binding proteins: parvalbumin, calmodulin, S100 proteins and calcineurin, calreticulin and calsequestrin [Yáñez M, Gil-Longo J, Campos-Toimil M. Calcium binding proteins. Adv Exp Med Biol. 2012;740:461-82. doi: 10.1007/978-94-007-2888-2_19. PMID: 22453954], [ Carapeto, A.P.; Marcuello, C.; Faísca, P.F.N.; Rodrigues, M.S. Morphological and Biophysical Study of S100A9 Protein Fibrils by Atomic Force Microscopy Imaging and Nanomechanical Analysis. Biomolecules 2024, 14, 1091].
Added
3) “It is known that the amplitude and shape of calcium impulse depend on the concentration of intracellular calcium-binding proteins (CBPs), which can delay or eliminate the effect of Ca2+ (…) Calretinin and calbinding belong to fast Ca2+-binding proteins (…) The action of such CBPs can be mimicked by using a synthetic analog, BAPTA” (lines 58-63). Did the authors also test the effect of calmoduling on the calcium intracellular signaling processes? This information should be placed in the respective Results sections.
Response: Calmodulin is not a buffer protein, but a regulatory one. Therefore, in this work, we did not study the effect of CaM. We studied the effect of CaM on the regulation of the Kv7 potassium channel, which is involved in the epileptiform activity [19]
4) Results. Did the authors carry out electrophysiology experiments in absence of calcium as negative control (e.g. with other positive divalent cations as magnesium or in presence of quelators)? Some further insights need to be furnished in this regard.
Response: The medium without Ca2+, suppresses the epileptiform rhythm. Ca2+ is needed to trigger and form a burst of AP and Ca2+ oscillations. It has been previously shown that when the input of Ca2+ is reduced, the amplitude and duration of the AP bursts decrease, and the Ca2+ pulse decreases accordingly. [Teplov I.Yu.,Tuleukhanov S.T., Zinchenko V.P. Regulation of action potential frequency and amplitude by T-type Ca2+ channel during spontaneous synchronous activity of hippocampal neurons. Biophysics, 2018, Vol. 63, No. 4, pp. 566–575.]
In other experiments, we used BAPTA-AM at concentrations of 10, 25, and 50 μM. In addition to BAPTA AM, we used EGTA-AM at concentrations of 25 μM, 50 μM, and Quin-2AM at a concentration of 30 μM. We observed a rapid attenuation of oscillations. These chelators had side effects due to their increased affinity for Ca2+. However, the qualitative effects were similar (see the figure below). We do not use a model without Mg2+, as it removes the NMDAR block.
|
Periodic neuronal Ca2+ pulse activity induced by the GABA(A) receptor inhibitor bicuculline (10 μM) in control. Before adding the intracellular Ca2+ buffer Quin-2 (30 μM). |
After 9 minutes of incubation with intracellular buffer Ca2+ Quin-2 (30 μM). In most glutamate neurons, the amplitude and duration of impulses decrease (gray-green curves). This corresponds to a reduction in the duration of AP bursts. In the minor population of GABAergic neurons, the amplitude and duration of impulses increase due to a decrease in pumping speed (red curves). |
5) “The mechanism (…) (lines 168-172). It was an issue related to the bibliography citation manager software tool. The authors should fix it. This comment needs to be taken into account for the rest of the main manuscript body text.
Response: Corrected
6) Finally, did the authors monitor the resting membrane potential and imput neuronal resistance during the Patch Clamp experiments to ensure that the cells remain healthy and viable after the calcium-activated currents?
Response: The resting membrane potential of GABAergic and glutamatergic neurons did not differ significantly and averaged -65 mV. The membrane potential at which the Na+ current was generated in GABAergic neurons was -55 mV on average and required a small depolarization to activate it. In contrast, glutamatergic neurons were activated at an average of -42 mV and required a strong depolarization. (inserted in the paragraph ”The effect of BAPTA on PDS clusters during epileptiform neuronal activity”. According to these parameters, the GABAergic neurons we selected coincide with low-threshold spiking (LTS) interneurons, which are also considered low-threshold and are activated by a small potential shift of -65 to 55 mV. In Figures 3A and 4A, the amplitude of the ПД did not change during the accumulation of BAPTA, indicating that the cells remained healthy and viable after the calcium-activated currents.
7) “Discussion” (lines 148-207). This section perfectly remarks the most relevant outcomes found by the authors in this work and also the promising future perspectives. It may be desirable to add a brief statement to discuss about the potential future action lines to pursue the topic covered in this research.
Response: In the future, we aim to control the rhythm of neural network development and recovery after damage. This work shows that intracellular fast Ca2+ binding proteins can be involved in this process. We will use the filling of Ca2+ buffers in inhibitory neurons to switch the epileptic rhythm into the rhythm of neural network recovery after brain injury. Inserted in the “Discussion”
When the buffers are filled, oscillations synchronize and the rhythm of epileptic activity develops. When the Ca2+ buffer is filled in GABAergic neurons containing CP-AMPA receptors, these neurons begin to secrete GABA and suppress other GABA neurons, thus removing their inhibitory effect and releasing the population of glutamate neurons that they innervate.
methods adequately described
Response: added
Results clearly presented Must be improved
Response: improved
Conclusions supported by the results Must be improved
Response: improved
all figures and tables clear and well-presented Must be improved
Response: improved
1) Introduction. “Generally, the term epileptiform activity of brain neurons (…) electrical activity is transformed into paraxysmal depolarizing shift (PDS) cluster (…) the brain neuron activity has been observed not only in epilepsy but also in stroke, Alzheimer’s disease (AD), Parkinson’s disease (PD) (…) early period of brain development” (lines 40-50). Could the authors provide quantitative data insights according to the worldwide global burdens of neuronological disorders as the above described? This will significantly aid the potential readers to better understand the significance of this devoted research.
Response:
According to the World Health Organization (WHO) tens of millions of people suffer from various neurological disorders worldwide. Approximately 50–70 million people have epilepsy [Liu J, Zhang P, Zou Q, Liang J, Chen Y, Cai Y, Li S, Li J, Su J, Li Q. Status of epilepsy in the tropics: An overlooked perspective. Epilepsia Open. 2023 Mar;8(1):32-45]. Parkinson’s disease affecting approximately 1% of the population over the age of 60 years, growing to 4% at the age of 80 years. [Polymeropoulos MH, Higgins JJ, Golbe LI, Johnson WG, Ide SE, Di Iorio G, Sanges G, Stenroos ES, Pho LT, Schaffer AA, Lazzarini AM, Nussbaum RL, Duvoisin RC. Mapping of a gene for Parkinson's disease to chromosome 4q21-q23. Science. 1996 Nov 15;274(5290):1197-9.] [Zaltieri M, Longhena F, Pizzi M, Missale C, Spano P, Bellucci A. Mitochondrial Dysfunction and α-Synuclein Synaptic Pathology in Parkinson's Disease: Who's on First? Parkinsons Dis. 2015;2015:108029. doi: 10.1155/2015/108029. Epub 2015 Mar 31. ]. Sixty-nine million individuals worldwide are estimated to sustain a Traumatic brain injury each year. [Dewan MC, Rattani A, Gupta S, Baticulon RE, Hung YC, Punchak M, Agrawal A, Adeleye AO, Shrime MG, Rubiano AM, Rosenfeld JV, Park KB. Estimating the global incidence of traumatic brain injury. J Neurosurg. 2018 Apr 27;130(4):1080-1097]. Stroke is the third highest cause of morbidity and mortality in many countries of the world, following ischemic heart disease and malignant diseases. [Petrović G. Faktori rizika u pojavi cerebrovaskularnog insulta [Risk factors for development of cerebrovascular stroke]. Med Pregl. 2000 Mar-Apr;53(3-4):207-14. Croatian. PMID: 10965691.] Approximately 1% of the global population (50–70 million people) have epilepsy [Ngugi, A.K.; Bottomley, C.; Kleinschmidt, I.; Sander, J.W.; Newton, C.R. Estimation of the burden of active and life-time epilepsy: A meta-analytic approach. Epilepsia 2010, 51, 883–890].
Response: Added
2) “Ca2+ ions are one of the main regulators of neuronal bursting activity (…) generation of AP burst during epileptiform activity is accompanied by Ca2+ impulses. (…) Ca2+ on transmitter release or on excitation” (lines 56-60). Here, even if I agree with these statements provided by the authors, it should be also remarkable to highlight the importance of intracellular calcium concentration not only in the onset and development of neurological disorders triggering the fibril formation [1], but also the impact of environmental conditions inner the brain [2]. This will strengthen the gathered findings in this research.
Response: Ca2+ ions regulate many processes in the cell. They do this both directly by binding to targets and indirectly by binding to Ca2+-binding proteins: parvalbumin, calmodulin, S100 proteins and calcineurin, calreticulin and calsequestrin [Yáñez M, Gil-Longo J, Campos-Toimil M. Calcium binding proteins. Adv Exp Med Biol. 2012;740:461-82. doi: 10.1007/978-94-007-2888-2_19. PMID: 22453954], [ Carapeto, A.P.; Marcuello, C.; Faísca, P.F.N.; Rodrigues, M.S. Morphological and Biophysical Study of S100A9 Protein Fibrils by Atomic Force Microscopy Imaging and Nanomechanical Analysis. Biomolecules 2024, 14, 1091].
Added
3) “It is known that the amplitude and shape of calcium impulse depend on the concentration of intracellular calcium-binding proteins (CBPs), which can delay or eliminate the effect of Ca2+ (…) Calretinin and calbinding belong to fast Ca2+-binding proteins (…) The action of such CBPs can be mimicked by using a synthetic analog, BAPTA” (lines 58-63). Did the authors also test the effect of calmoduling on the calcium intracellular signaling processes? This information should be placed in the respective Results sections.
Response: Calmodulin is not a buffer protein, but a regulatory one. Therefore, in this work, we did not study the effect of CaM. We studied the effect of CaM on the regulation of the Kv7 potassium channel, which is involved in the epileptiform activity [19]
4) Results. Did the authors carry out electrophysiology experiments in absence of calcium as negative control (e.g. with other positive divalent cations as magnesium or in presence of quelators)? Some further insights need to be furnished in this regard.
Response: The medium without Ca2+, suppresses the epileptiform rhythm. Ca2+ is needed to trigger and form a burst of AP and Ca2+ oscillations. It has been previously shown that when the input of Ca2+ is reduced, the amplitude and duration of the AP bursts decrease, and the Ca2+ pulse decreases accordingly. [Teplov I.Yu.,Tuleukhanov S.T., Zinchenko V.P. Regulation of action potential frequency and amplitude by T-type Ca2+ channel during spontaneous synchronous activity of hippocampal neurons. Biophysics, 2018, Vol. 63, No. 4, pp. 566–575.]
In other experiments, we used BAPTA-AM at concentrations of 10, 25, and 50 μM. In addition to BAPTA AM, we used EGTA-AM at concentrations of 25 μM, 50 μM, and Quin-2AM at a concentration of 30 μM. We observed a rapid attenuation of oscillations. These chelators had side effects due to their increased affinity for Ca2+. However, the qualitative effects were similar (see the figure below). We do not use a model without Mg2+, as it removes the NMDAR block.
|
Periodic neuronal Ca2+ pulse activity induced by the GABA(A) receptor inhibitor bicuculline (10 μM) in control. Before adding the intracellular Ca2+ buffer Quin-2 (30 μM). |
After 9 minutes of incubation with intracellular buffer Ca2+ Quin-2 (30 μM). In most glutamate neurons, the amplitude and duration of impulses decrease (gray-green curves). This corresponds to a reduction in the duration of AP bursts. In the minor population of GABAergic neurons, the amplitude and duration of impulses increase due to a decrease in pumping speed (red curves). |
5) “The mechanism (…) (lines 168-172). It was an issue related to the bibliography citation manager software tool. The authors should fix it. This comment needs to be taken into account for the rest of the main manuscript body text.
Response: Corrected
6) Finally, did the authors monitor the resting membrane potential and imput neuronal resistance during the Patch Clamp experiments to ensure that the cells remain healthy and viable after the calcium-activated currents?
Response: The resting membrane potential of GABAergic and glutamatergic neurons did not differ significantly and averaged -65 mV. The membrane potential at which the Na+ current was generated in GABAergic neurons was -55 mV on average and required a small depolarization to activate it. In contrast, glutamatergic neurons were activated at an average of -42 mV and required a strong depolarization. (inserted in the paragraph ”The effect of BAPTA on PDS clusters during epileptiform neuronal activity”. According to these parameters, the GABAergic neurons we selected coincide with low-threshold spiking (LTS) interneurons, which are also considered low-threshold and are activated by a small potential shift of -65 to 55 mV. In Figures 3A and 4A, the amplitude of the ПД did not change during the accumulation of BAPTA, indicating that the cells remained healthy and viable after the calcium-activated currents.
7) “Discussion” (lines 148-207). This section perfectly remarks the most relevant outcomes found by the authors in this work and also the promising future perspectives. It may be desirable to add a brief statement to discuss about the potential future action lines to pursue the topic covered in this research.
Response: In the future, we aim to control the rhythm of neural network development and recovery after damage. This work shows that intracellular fast Ca2+ binding proteins can be involved in this process. We will use the filling of Ca2+ buffers in inhibitory neurons to switch the epileptic rhythm into the rhythm of neural network recovery after brain injury. Inserted in the “Discussion”
When the buffers are filled, oscillations synchronize and the rhythm of epileptic activity develops. When the Ca2+ buffer is filled in GABAergic neurons containing CP-AMPA receptors, these neurons begin to secrete GABA and suppress other GABA neurons, thus removing their inhibitory effect and releasing the population of glutamate neurons that they innervate.
methods adequately described
Response: added
Results clearly presented Must be improved
Response: improved
Conclusions supported by the results Must be improved
Response: improved
all figures and tables clear and well-presented Must be improved
Response: improved
Reviewer 2 Report
Comments and Suggestions for Authors
Please provide statistical analyses of the data from multiple experiments to make conclusions.
Authors should perform dose response experiments with BAPTA-AM.
Figures should be arranged better. Especially, “Figure 3A, 4A” is confusing.
Methods section, first paragraph: What does “Error! Reference source not found.” mean?
From the Methods section, it is not clear what animals were used. Please describe species, sex, age/weight of the animals.
Author Response
- Please provide statistical analyses of the data from multiple experiments to make conclusions.
Response: added
- Authors should perform dose response experiments with BAPTA-AM.
Response: The concentration of BAPTA AM was selected experimentally. In other experiments, BAPTA was used at concentrations of 10, 25, and 50 μM. Lower concentrations did not affect the pulse parameters, while higher concentrations caused rapid inhibition of pulsations. The relationship between the dose and the effect of intracellular BAPTA is evident when BAPTA is accumulated during the experiment.
Response: added in “Introduction”
- Figures should be arranged better. Especially, “Figure 3A, 4A” is confusing.
Response: improved
Methods section, first paragraph: What does “Error! Reference source not found.” mean?
Response: references have been sorted and changed
From the Methods section, it is not clear what animals were used. Please describe species, sex, age/weight of the animals.
Response: Neuron-glial cultures were derived from the hippocampi of newborn Wistar rats (P0–2), as described earlier (Laryushkin et al., 2023; Gaidin et al., 2023). Wistar pups (P0-2) of both sexes were euthanized with deep-inhaled anesthesia and decapitated. (Оriginal source—animal facility of The Branch of the IBCh RAS in Pushchino; RRID: Not registered).
Response: inserted into the text
- Please provide statistical analyses of the data from multiple experiments to make conclusions.
Response: added
- Authors should perform dose response experiments with BAPTA-AM.
Response: The concentration of BAPTA AM was selected experimentally. In other experiments, BAPTA was used at concentrations of 10, 25, and 50 μM. Lower concentrations did not affect the pulse parameters, while higher concentrations caused rapid inhibition of pulsations. The relationship between the dose and the effect of intracellular BAPTA is evident when BAPTA is accumulated during the experiment.
Response: added in “Introduction”
- Figures should be arranged better. Especially, “Figure 3A, 4A” is confusing.
Response: improved
Methods section, first paragraph: What does “Error! Reference source not found.” mean?
Response: references have been sorted and changed
From the Methods section, it is not clear what animals were used. Please describe species, sex, age/weight of the animals.
Response: Neuron-glial cultures were derived from the hippocampi of newborn Wistar rats (P0–2), as described earlier (Laryushkin et al., 2023; Gaidin et al., 2023). Wistar pups (P0-2) of both sexes were euthanized with deep-inhaled anesthesia and decapitated. (Оriginal source—animal facility of The Branch of the IBCh RAS in Pushchino; RRID: Not registered).
Response: inserted into the text
- Please provide statistical analyses of the data from multiple experiments to make conclusions.
Response: added
- Authors should perform dose response experiments with BAPTA-AM.
Response: The concentration of BAPTA AM was selected experimentally. In other experiments, BAPTA was used at concentrations of 10, 25, and 50 μM. Lower concentrations did not affect the pulse parameters, while higher concentrations caused rapid inhibition of pulsations. The relationship between the dose and the effect of intracellular BAPTA is evident when BAPTA is accumulated during the experiment.
Response: added in “Introduction”
- Figures should be arranged better. Especially, “Figure 3A, 4A” is confusing.
Response: improved
Methods section, first paragraph: What does “Error! Reference source not found.” mean?
Response: references have been sorted and changed
From the Methods section, it is not clear what animals were used. Please describe species, sex, age/weight of the animals.
Response: Neuron-glial cultures were derived from the hippocampi of newborn Wistar rats (P0–2), as described earlier (Laryushkin et al., 2023; Gaidin et al., 2023). Wistar pups (P0-2) of both sexes were euthanized with deep-inhaled anesthesia and decapitated. (Оriginal source—animal facility of The Branch of the IBCh RAS in Pushchino; RRID: Not registered).
Response: inserted into the text
- Please provide statistical analyses of the data from multiple experiments to make conclusions.
Response: added
- Authors should perform dose response experiments with BAPTA-AM.
Response: The concentration of BAPTA AM was selected experimentally. In other experiments, BAPTA was used at concentrations of 10, 25, and 50 μM. Lower concentrations did not affect the pulse parameters, while higher concentrations caused rapid inhibition of pulsations. The relationship between the dose and the effect of intracellular BAPTA is evident when BAPTA is accumulated during the experiment.
Response: added in “Introduction”
- Figures should be arranged better. Especially, “Figure 3A, 4A” is confusing.
Response: improved
Methods section, first paragraph: What does “Error! Reference source not found.” mean?
Response: references have been sorted and changed
From the Methods section, it is not clear what animals were used. Please describe species, sex, age/weight of the animals.
Response: Neuron-glial cultures were derived from the hippocampi of newborn Wistar rats (P0–2), as described earlier (Laryushkin et al., 2023; Gaidin et al., 2023). Wistar pups (P0-2) of both sexes were euthanized with deep-inhaled anesthesia and decapitated. (Оriginal source—animal facility of The Branch of the IBCh RAS in Pushchino; RRID: Not registered).
Response: inserted into the text
- Please provide statistical analyses of the data from multiple experiments to make conclusions.
Response: added
- Authors should perform dose response experiments with BAPTA-AM.
Response: The concentration of BAPTA AM was selected experimentally. In other experiments, BAPTA was used at concentrations of 10, 25, and 50 μM. Lower concentrations did not affect the pulse parameters, while higher concentrations caused rapid inhibition of pulsations. The relationship between the dose and the effect of intracellular BAPTA is evident when BAPTA is accumulated during the experiment.
Response: added in “Introduction”
- Figures should be arranged better. Especially, “Figure 3A, 4A” is confusing.
Response: improved
Methods section, first paragraph: What does “Error! Reference source not found.” mean?
Response: references have been sorted and changed
From the Methods section, it is not clear what animals were used. Please describe species, sex, age/weight of the animals.
Response: Neuron-glial cultures were derived from the hippocampi of newborn Wistar rats (P0–2), as described earlier (Laryushkin et al., 2023; Gaidin et al., 2023). Wistar pups (P0-2) of both sexes were euthanized with deep-inhaled anesthesia and decapitated. (Оriginal source—animal facility of The Branch of the IBCh RAS in Pushchino; RRID: Not registered).
Response: inserted into the text
- Please provide statistical analyses of the data from multiple experiments to make conclusions.
Response: added
- Authors should perform dose response experiments with BAPTA-AM.
Response: The concentration of BAPTA AM was selected experimentally. In other experiments, BAPTA was used at concentrations of 10, 25, and 50 μM. Lower concentrations did not affect the pulse parameters, while higher concentrations caused rapid inhibition of pulsations. The relationship between the dose and the effect of intracellular BAPTA is evident when BAPTA is accumulated during the experiment.
Response: added in “Introduction”
- Figures should be arranged better. Especially, “Figure 3A, 4A” is confusing.
Response: improved
Methods section, first paragraph: What does “Error! Reference source not found.” mean?
Response: references have been sorted and changed
From the Methods section, it is not clear what animals were used. Please describe species, sex, age/weight of the animals.
Response: Neuron-glial cultures were derived from the hippocampi of newborn Wistar rats (P0–2), as described earlier (Laryushkin et al., 2023; Gaidin et al., 2023). Wistar pups (P0-2) of both sexes were euthanized with deep-inhaled anesthesia and decapitated. (Оriginal source—animal facility of The Branch of the IBCh RAS in Pushchino; RRID: Not registered).
Response: inserted into the text
- Please provide statistical analyses of the data from multiple experiments to make conclusions.
Response: added
- Authors should perform dose response experiments with BAPTA-AM.
Response: The concentration of BAPTA AM was selected experimentally. In other experiments, BAPTA was used at concentrations of 10, 25, and 50 μM. Lower concentrations did not affect the pulse parameters, while higher concentrations caused rapid inhibition of pulsations. The relationship between the dose and the effect of intracellular BAPTA is evident when BAPTA is accumulated during the experiment.
Response: added in “Introduction”
- Figures should be arranged better. Especially, “Figure 3A, 4A” is confusing.
Response: improved
Methods section, first paragraph: What does “Error! Reference source not found.” mean?
Response: references have been sorted and changed
From the Methods section, it is not clear what animals were used. Please describe species, sex, age/weight of the animals.
Response: Neuron-glial cultures were derived from the hippocampi of newborn Wistar rats (P0–2), as described earlier (Laryushkin et al., 2023; Gaidin et al., 2023). Wistar pups (P0-2) of both sexes were euthanized with deep-inhaled anesthesia and decapitated. (Оriginal source—animal facility of The Branch of the IBCh RAS in Pushchino; RRID: Not registered).
Response: inserted into the text
- Please provide statistical analyses of the data from multiple experiments to make conclusions.
Response: added
- Authors should perform dose response experiments with BAPTA-AM.
Response: The concentration of BAPTA AM was selected experimentally. In other experiments, BAPTA was used at concentrations of 10, 25, and 50 μM. Lower concentrations did not affect the pulse parameters, while higher concentrations caused rapid inhibition of pulsations. The relationship between the dose and the effect of intracellular BAPTA is evident when BAPTA is accumulated during the experiment.
Response: added in “Introduction”
- Figures should be arranged better. Especially, “Figure 3A, 4A” is confusing.
Response: improved
Methods section, first paragraph: What does “Error! Reference source not found.” mean?
Response: references have been sorted and changed
From the Methods section, it is not clear what animals were used. Please describe species, sex, age/weight of the animals.
Response: Neuron-glial cultures were derived from the hippocampi of newborn Wistar rats (P0–2), as described earlier (Laryushkin et al., 2023; Gaidin et al., 2023). Wistar pups (P0-2) of both sexes were euthanized with deep-inhaled anesthesia and decapitated. (Оriginal source—animal facility of The Branch of the IBCh RAS in Pushchino; RRID: Not registered).
Response: inserted into the text
- Please provide statistical analyses of the data from multiple experiments to make conclusions.
Response: added
- Authors should perform dose response experiments with BAPTA-AM.
Response: The concentration of BAPTA AM was selected experimentally. In other experiments, BAPTA was used at concentrations of 10, 25, and 50 μM. Lower concentrations did not affect the pulse parameters, while higher concentrations caused rapid inhibition of pulsations. The relationship between the dose and the effect of intracellular BAPTA is evident when BAPTA is accumulated during the experiment.
Response: added in “Introduction”
- Figures should be arranged better. Especially, “Figure 3A, 4A” is confusing.
Response: improved
Methods section, first paragraph: What does “Error! Reference source not found.” mean?
Response: references have been sorted and changed
From the Methods section, it is not clear what animals were used. Please describe species, sex, age/weight of the animals.
Response: Neuron-glial cultures were derived from the hippocampi of newborn Wistar rats (P0–2), as described earlier (Laryushkin et al., 2023; Gaidin et al., 2023). Wistar pups (P0-2) of both sexes were euthanized with deep-inhaled anesthesia and decapitated. (Оriginal source—animal facility of The Branch of the IBCh RAS in Pushchino; RRID: Not registered).
Response: inserted into the text
- Please provide statistical analyses of the data from multiple experiments to make conclusions.
Response: added
- Authors should perform dose response experiments with BAPTA-AM.
Response: The concentration of BAPTA AM was selected experimentally. In other experiments, BAPTA was used at concentrations of 10, 25, and 50 μM. Lower concentrations did not affect the pulse parameters, while higher concentrations caused rapid inhibition of pulsations. The relationship between the dose and the effect of intracellular BAPTA is evident when BAPTA is accumulated during the experiment.
Response: added in “Introduction”
- Figures should be arranged better. Especially, “Figure 3A, 4A” is confusing.
Response: improved
Methods section, first paragraph: What does “Error! Reference source not found.” mean?
Response: references have been sorted and changed
From the Methods section, it is not clear what animals were used. Please describe species, sex, age/weight of the animals.
Response: Neuron-glial cultures were derived from the hippocampi of newborn Wistar rats (P0–2), as described earlier (Laryushkin et al., 2023; Gaidin et al., 2023). Wistar pups (P0-2) of both sexes were euthanized with deep-inhaled anesthesia and decapitated. (Оriginal source—animal facility of The Branch of the IBCh RAS in Pushchino; RRID: Not registered).
Response: inserted into the text
Round 2
Reviewer 1 Report
Comments and Suggestions for Authors
The authors did a great deal of effort to cover all the raised suggestions by the Reviewers. For this reason the scientific manuscript quality was greatly improved. Based on the novelty and significance of the gathered results, I warmly endorse this work for further publication in the International Journal of Molecular Sciences in the present form.
Reviewer 2 Report
Comments and Suggestions for Authors.